# Effect of Forward and Reverse Suturing on Nerve Regeneration Following Facial Nerve Axotomy

**DOI:** 10.3390/biomedicines12112531

**Published:** 2024-11-05

**Authors:** Jae Min Lee, Jong Woo Chung, Na Young Jeong, Junyang Jung, Sung Soo Kim, Su Jin Jeong, Seung Geun Yeo

**Affiliations:** 1Department of Otorhinolaryngology—Head and Neck Surgery, College of Medicine, Kyung Hee University Medical Center, Seoul 02447, Republic of Korea; sunjaesa@hanmail.net; 2Department of Otorhinolaryngology—Head and Neck Surgery, Asan Medical Center, College of Medicine, University of Ulsan, Seoul 02447, Republic of Korea; jwchung@amc.seoul.kr; 3Department of Anatomy and Cell Biology, College of Medicine, Dong-A University, Busan 49201, Republic of Korea; jnyjjy@dau.ac.kr; 4Department of Anatomy and Neurobiology, College of Medicine, Kyung Hee University, Seoul 02447, Republic of Korea; jjung@khu.ac.kr; 5Department of Biochemistry and Molecular Biology, College of Medicine, Kyung Hee University, Seoul 02447, Republic of Korea; sgskim@khu.ac.kr; 6Clinical Research Institute, Kyung Hee University Medical Center, Seoul 02447, Republic of Korea; rainofsujin@naver.com; 7Department of Convergence Medicine, College of Medicine, Kyung Hee University, Seoul 02447, Republic of Korea

**Keywords:** facial nerve injury, axotomy, nerve regeneration, forward suture, reverse suture

## Abstract

Background/Objectives: When the facial nerve is severed and a nerve graft is required, motor nerves are typically connected in the forward direction, while sensory nerves are connected in the reverse direction. However, there is limited research on the effects of reversing this connection, and no studies have been conducted using the same facial nerve. This study aimed to investigate the effects of forward and reverse suturing on nerve regeneration following facial nerve axotomy. Methods: The facial nerve trunk of male Sprague Dawley rats was incised to induce facial nerve injury, and autografts were sutured using both forward and reverse methods. Behavioral tests, including whisker reflex and eye blink tests, were conducted. Histological analyses, including toluidine blue staining and transmission electron microscopy (TEM), were performed to evaluate axon recovery. Results: Behavioral experiments showed signs of recovery at 3–4 weeks in both the forward and reverse suture groups, with no significant differences between the two methods (*p* < 0.01). Histological analysis showed partial recovery by 8 weeks in both groups. Toluidine blue staining indicated a reduction in the number of axons at 4 weeks, with partial recovery at 8 weeks (*p* < 0.001) in both groups. TEM analysis revealed that myelin fiber thickness was restored in both the forward and reverse suture groups, though it remained thinner compared to normal (*p* < 0.01). Conclusions: Our results suggest that the direction of nerve suturing (forward vs. reverse) does not significantly impact nerve regeneration or functional recovery. Both suturing methods demonstrated similar recovery effects, with no significant differences in microstructural regeneration. Future studies should investigate the molecular mechanisms underlying nerve regeneration and extend the observation period to provide a more comprehensive understanding of this process.

## 1. Introduction

Peripheral nerves are susceptible to compression, laceration, or other forms of trauma, making them vulnerable to damage from various injuries and diseases. Facial nerve damage is particularly unique because it affects both functional and emotional aspects of the face, leading to significant emotional distress and impairing social and economic activities [1]. Symptoms of facial nerve injury include difficulty closing the affected eyelid, loss of normal nasolabial folds, and asymmetrical facial expressions during crying. Facial nerve paralysis impairs muscles involved in smiling, blinking, and other facial movements, hindering the ability to express emotions [2]. The causes of facial nerve damage vary, including idiopathic Bell’s palsy, viral infections, skull fractures from traffic accidents or falls, facial nerve compression, and complications from otitis media [3]. Trauma from traffic accidents and falls frequently leads to facial nerve injury, with the risk being especially high when accompanied by skull fractures [4]. These injuries can result in more than nerve compression; they may cause axotomy or severe nerve damage. Facial nerve axotomy, where nerve fibers are severed, complicates regeneration and limits the recovery of motor function. The facial nerve is most commonly injured where it exits the skull through the stylomastoid foramen, leading to deficits similar to those seen in lower motor neuron lesions [4,5].

Complete nerve damage due to axotomy is particularly serious, and even partial recovery can be difficult [1]. In such cases, conservative treatments are often inadequate, and surgical intervention, such as nerve reconstruction, becomes necessary [6]. The outcomes of surgical treatments vary depending on the extent of injury, but achieving full recovery is challenging, particularly when the nerve is completely severed [7]. For peripheral nerve injuries, nerve grafting and suturing are commonly employed to repair damage [8]. When direct suturing is not feasible, nerve grafting helps restore nerve function by providing a pathway for regeneration. Successful nerve reconstruction is essential for restoring facial function and significantly improving the patient’s quality of life.

The orientation of the autologous nerve graft can significantly affect nerve regeneration [9]. In the forward suture technique, the distal and proximal nerve ends are aligned in the same direction, facilitating accurate nerve signal transmission and potentially leading to more effective functional recovery [10]. Conversely, the reverse suture technique connects the nerve ends in opposite directions, reversing the nerve fibers [11]. Although reverse suturing does not directly contribute to nerve regeneration in some cases, it can still provide a pathway for functional recovery [12]. One study suggested that nerve reversal in grafts does not significantly affect functional outcomes [13]. This reversal can introduce confusion into the nerve regeneration process, but it still provides a pathway for the damaged nerve to recover function. Therefore, the purpose of this study was to compare the effects of the forward suture method, which aligns the nerve fibers according to their original direction, with the reverse suture method, which connects them in the opposite direction, on nerve recovery following facial nerve axotomy. While previous research has primarily focused on sensory nerves, this study investigates the facial nerve, a motor nerve, to observe and compare outcomes between the two suturing techniques.

## 2. Materials and Methods

### 2.1. Animals

Male Sprague Dawley rats, aged 6–7 weeks and weighing 200–250 g, were procured from Orient Bio (Seongnam, Gyeonggi-do, Republic of Korea). The animals were housed in an environment-controlled facility maintained at 22 ± 2 °C with 50% humidity and a 12 h light/dark cycle. Rats had unrestricted access to food and water throughout the study. Following a 7-day acclimation period, the animals were randomly assigned to either the experimental axotomy group (*n* = 18) or the sham group (*n* = 3). The axotomy group was divided into control (*n* = 6), forward suture (*n* = 6), and reverse suture (*n* = 6) groups.

### 2.2. Forward and Reverse Suturing Procedures After Facial Nerve Axotomy

Prior to surgery, each rat was fasted for 12 h to reduce the risk of aspiration during anesthesia. Anesthesia was induced in an induction chamber using 5% isoflurane (Foran solution, Hwaseong Joongwae, Republic of Korea) in 80% oxygen and maintained via a nose cone at 3% isoflurane. The rat’s head was shaved and the surgical site was prepared with povidone–iodine solution followed by 70% ethanol. Body temperature was maintained at 37 °C using a heating pad throughout the procedure. A 2 cm left postauricular incision was made to expose the facial nerve anatomy. In the axotomy group, the main trunk of the facial nerve was carefully isolated from surrounding tissues using microsurgical techniques. A 5 mm segment of the facial nerve trunk was completely transected using microsurgical scissors, with care taken to avoid stretching or crushing the nerve ends. The excised nerve segment was immediately placed in sterile 1× phosphate-buffered saline (PBS) solution at 4 °C for temporary preservation. The contralateral facial nerve underwent an identical procedure to serve as an internal control.

The preserved nerve segment was then reattached using 10-0 ETHILON™ nylon sutures (Black, Somerville, NJ, USA) under a surgical microscope (Zeiss OPMI^®^ VARIO/S88, Carl Zeiss Meditec AG, Jena, Germany) at 40× magnification. Post-suturing, the surgical site was irrigated with sterile saline, and 3-0 nylon sutures were applied to the skin. Sham group rats underwent an identical surgical approach, including the postauricular incision and exposure of the facial nerve, but without nerve transection. The incision was closed in the same manner as the experimental group. Post-operatively, rats were monitored in a recovery cage under a heat lamp until fully conscious. The animals were closely monitored for signs of distress, as well as changes in weight, behavior, and wound conditions for 7 days post-surgery. Skin sutures were removed on postoperative day 7.

### 2.3. Behavioral Tests

Facial nerve function was assessed at weeks 1, 2, 3, and 4 post-surgery by two blinded observers to minimize bias. The whisker movement test, used to evaluate whisker muscle function, involved securely holding the rat. Whisker movement was evaluated using a modified five-point Vibrissae Observation Scale with the following scoring system: 5 points: normal symmetric forward movement; 4 points: normal movement with slight backward tilt (≤30°); 3 points: significant movement with moderate backward tilt (30–60°); 2 points: slight movement with severe backward tilt (>60°); 1 point: no detectable movement. Eyelid function was tested using controlled air puff stimulation delivered by a calibrated air pump at a distance of 1 cm from the cornea. The blink reflex was scored on the Eye Closing and Blinking Reflex Observation Scale as follows: 5 points: complete, rapid eye closure; 4 points: 75% closure with normal blink speed; 3 points: 50% closure with slowed blink response; 2 points: visible muscle contraction without eyelid closure; 1 point: no detectable movement. Each test was performed three times per session with a 30 s interval between trials. The median score of the three trials was recorded for analysis.

### 2.4. Toluidine Blue Staining of Myelinated Fibers

In the axotomy group, rats were divided into the control group (*n* = 3), the forward suture group (*n* = 3), and the reverse suture group (*n* = 3) at 4 and 8 weeks (*n* = 3 per group at each time point). They were anesthetized with ether and their facial nerves were carefully extracted. The facial nerve was carefully dissected and a 5 mm segment was harvested 2 mm distal to the suture site. Samples were immediately fixed in a solution containing 2% glutaraldehyde (Electron Microscopy Sciences, Hatfield, PA, USA), 2% paraformaldehyde (Wako Pure Chemical, Osaka, Japan), and 0.1 mol/L cacodylate buffer (pH 7.4) for 24 h at 4 °C. Post-fixation was performed in 2% osmium tetroxide in 0.1 mol/L cacodylate buffer for 2 h at room temperature. Following fixation, the specimens were dehydrated through a graded ethanol series (50%, 70%, 80%, 90%, 95%, and 100%) and embedded in Quetol-812 resin (Nisshin EM, Tokyo, Japan). The resin blocks were polymerized at 60 °C for 48 h. Semi-thin sections (1 μm thickness) were cut using an ultramicrotome (Leica EM UC7, Leica Microsystems, Wetzlar, Germany) with a glass knife and stained with 0.5% toluidine blue for 30 s at 60 °C. Stained sections were examined using a NanoZoomer 2.0-HT scanner (Hamamatsu Photonics, Shizuoka, Japan) at 40× magnification. Quantitative analysis was performed using NDP.view 1.2.25 software. In the analysis of toluidine blue-stained sections, the following parameters were measured across the entire cross-section of the regenerated nerve. Specifically, two independent observers, both of whom have extensive experience in histological analysis (more than 3 years of experience), assessed the samples. Each observer measured three non-overlapping fields of view (FOV) per section to ensure a comprehensive evaluation. The FOV for each analysis was standardized with a specified magnification or size, allowing for consistent measurements across all samples. The average values for each parameter were calculated from these three fields to enhance reliability. The observers were blinded to the experimental groups to eliminate bias in their assessments.

### 2.5. Transmission Electron Microscopy of the Regenerated Nerves

In the axotomy group, rats were divided into the control group (*n* = 3), the forward suture group (*n* = 3), and the reverse suture group (*n* = 3) at 4 and 8 weeks (*n* = 3 per group at each time point). They were anesthetized with ether and their facial nerves were carefully extracted. For transmission electron microscopy, ultrathin sections (70 nm thickness) were prepared from the same resin blocks used for light microscopy. Sections were collected on EM fine-grid F-200 copper grids (Nisshin) and double-stained with 2% uranyl acetate for 15 min followed by lead citrate solution (Sigma-Aldrich, St. Louis, MO, USA) for 5 min. Stained sections were examined under a JEM1200EX transmission electron microscope (JEOL, Tokyo, Japan) operated at an accelerating voltage of 80 kV. Digital images were captured using a CCD camera (Olympus Soft Imaging Solutions, Münster, Germany) at magnifications ranging from 5000× to 20,000×. The axon diameter was measured as the total diameter of the axon, including the myelin and axon sheath, using ImageJ software (https://imagej.net/ij/) (National Institutes of Health, Bethesda, MD, USA). Myelin and axon thickness were calculated as the difference between the outer and inner diameter of the myelin and axon sheath divided by two. The g-ratio was calculated to assess myelin integrity by dividing the inner axonal diameter by the outer diameter of the myelinated axon sheath. Both myelin and axon thicknesses were determined from the TEM images, with the myelin thickness calculated as half of the difference between the outer and inner diameters of the myelinated axon.

### 2.6. Statistical Analysis

The data are presented as means ± standard error of the mean (SEM) and represent results from at least two independent replicates. Statistical analyses were performed using SPSS software (version 25; IBM SPSS Corp., Armonk, NY, USA). To evaluate differences among groups, a one-way ANOVA was conducted, followed by Tukey’s post hoc test for pairwise comparisons. To analyze the interaction effects by week, a general linear model (GLM) was employed to assess both the group and time-point interactions comprehensively. Statistical significance was defined as *p* < 0.05 for all analyses.

## 3. Results

### 3.1. Whisker Movement and Eyelid Blink Reflex Tests

The results of the eyelid blink reflex and whisker movement tests, performed at 1, 2, 3, and 4 weeks following the axotomy-induced facial nerve injury, are shown in Figure 1. In the whisker movement (vibrissae muscle) test (Figure 1A), there were no significant differences between the forward and reverse suture groups compared to the control group at 1 week (*p* = 0.127) and 2 weeks (*p* = 0.094) post-injury. However, by 4 weeks (*p* < 0.001), both the forward and reverse suture groups showed significant recovery in facial nerve function compared to the control group. Similarly, in the eyelid blink reflex test (Figure 1B), no significant differences were observed between the suture groups and the control group at 1 week (*p* = 0.127), 2 weeks (*p* = 0.238), and 3 weeks (*p* = 0.066) post-injury. However, by 4 weeks (*p* < 0.01), both the forward and reverse suture groups exhibited significantly improved facial nerve function compared to the control group.

### 3.2. Regenerated Nerve Morphology with Forward and Reverse Sutures After Facial Nerve Injury

The morphological characteristics of the regenerated facial nerves at 4 and 8 weeks post-axotomy were assessed in rats subjected to forward and reverse suture techniques, visualized using toluidine blue staining (Figure 2A). At 4 weeks post-injury, both suture groups (forward and reverse) showed markedly more severe nerve damage than the sham group (F = 40.619, *p* < 0.001) (Figure 2B). By 8 weeks post-injury, however, axonal counts in the forward and reverse suture groups were not significantly different from those in the sham group (F = 2.337, *p* = 0.118) (Figure 2C). At 8 weeks post-injury, both suture groups demonstrated morphological recovery of the nerves, with an increase in the number of axons observed.

### 3.3. Morphology of Myelin Sheath with Forward and Reverse Sutures After Facial Nerve Injury

The results illustrate the morphology of the regenerated myelin and axon sheaths in rats at 8 weeks post-facial nerve axotomy, as observed through transmission electron microscopy (TEM) (Figure 3A). The g-ratios in both the forward and reverse suture groups were significantly lower than those in the sham group (F = 5.823, *p* < 0.001) (Figure 3B).At 8 weeks, the thickness of the myelin sheath significantly decreased in comparison to the sham group (F = 10.917, *p* < 0.001) (Figure 3C), and similarly, axon sheath thickness was also reduced compared to the sham group (F = 21.228, *p* < 0.001) (Figure 3D). The g-ratio analysis, used to assess myelin thickness relative to axon diameter, quantitatively measured the extent of myelination recovery. Neither the forward nor reverse suturing techniques improved nerve recovery at 8 weeks, as assessed by TEM analysis, and there was no significant directional effect of suture type on nerve regeneration outcomes.

## 4. Discussion

Our previous results indicated that facial nerve function, as measured by blink and whisker reflexes, did not recover even 12 weeks after axotomy, suggesting a severe and long-term impairment of facial nerve function [14,15]. When we re-harvested the contralateral facial nerve and sutured the severed facial nerve in both forward and reverse directions, signs of recovery were observed 3 to 4 weeks later compared to the control group (facial nerve axotomy). However, the behavioral tests showed no significant difference between the forward and reverse suturing methods. Toluidine blue staining, used to assess nerve regeneration at the tissue level, revealed a reduction in the number of axons in both the forward and reverse suture groups at 4 weeks compared to the sham group. Although behavioral tests indicated some recovery of facial nerve function at 3 to 4 weeks, the number of regenerated axons had not fully recovered.

We assessed the effects of forward and reverse suturing on nerve regeneration using toluidine blue staining and transmission electron microscopy (TEM) analysis. The behavioral tests indicated recovery of facial nerve function after 4 weeks, but TEM analysis confirmed that regeneration was not fully restored, both structurally and functionally. TEM evaluation, which examines detailed ultrastructural changes in nerve regeneration, indicated reduced precise regeneration of myelinated fibers and axon conditions in both the forward and reverse suture groups. Both suturing techniques demonstrated normal nerve regeneration without microstructural issues. By 8 weeks, both the forward and reverse suture groups showed partial recovery in axon numbers compared to the sham group, with no significant differences between the two methods. Our findings align with previous studies suggesting no significant difference in regeneration outcomes between forward and reverse grafts [13].

Peripheral nerve repair for complete amputation injuries often employs reconstructive techniques requiring suturing. When the facial nerve is severed, nerve transfer procedures are used to connect the sural nerve directly to the facial nerve in an attempt to restore function [16]. The most effective reconstructive techniques include direct nerve anastomosis and nerve grafting. The recent literature on approaches and techniques for repairing facial nerve injuries continues to emphasize surgical methods as critical for functional recovery [17]. In a clinical study using cadaveric heads, the sural nerve and great auricular nerve were grafted to treat facial dysfunction, but complications may occur [18]. Autologous nerve grafts, taken from the patient’s own body, are commonly used to repair damaged nerves [19,20]. While effective in bridging significant nerve gaps and avoiding immune rejection, their use is limited by factors such as tissue availability, the need for a second surgical procedure to harvest the graft, and functional loss at the donor site [21,22]. Surgical repairs using auricular nerve grafts, sural nerve grafts, and hypoglossal nerve–facial nerve anastomosis have proven effective in restoring facial nerve function [7]. Facial nerve regeneration was improved in patients with facial nerve paralysis through the use of hypoglossal–facial anastomosis [23]. After performing hypoglossal–facial anastomosis due to facial nerve transection, there was a significant improvement in facial nerve function 24 months post-surgery [24].

The orientation of nerve grafts—whether in their normal or reversed polarity—significantly affects axonal regeneration in peripheral nerves. Reversed-polarity grafts can be more effective in guiding the regeneration of axons and may reduce the risk of misdirected axonal regeneration, which is a potential complication associated with normal polarity grafts [25]. Additionally, reverse unilateral nerve repair—where the distal stump of a severed nerve is sutured in reverse to the intact nerve—has been shown to accelerate nerve regeneration [26]. This technique holds practical advantages for peripheral nerve injuries, as it helps prevent the misalignment of nerve fibers. Reversed-polarity grafts may also decrease the likelihood of misdirected axonal sprouting, thereby enhancing distal regeneration [27]. In a study evaluating the effects of nerve graft polarity on regeneration and function, the sciatic nerve of male Sprague Dawley rats was severed and sutured while either maintaining the original polarity or applying reversed polarity. The findings indicated no significant difference in nerve regeneration between the two methods [28]. Another investigation into the proximo-distal orientation of peripheral nerve grafts revealed that while regeneration occurred in both orientations, the reversed grafts exhibited less loss in cross-sectional area compared to normally oriented grafts [25]. Furthermore, research on nerve regeneration following sciatic nerve transection, which involved forward and reverse suturing (with polarity reversed by 180 degrees), indicated no difference in outcomes regardless of the reversal or rotation [29]. Similarly, in a study involving the bilateral transection of the common peroneal nerve in rabbits, suturing in either the original or reversed direction did not affect nerve regeneration, as assessed through electrophysiological and histological methods [30]. However, another evaluation of axonal regeneration through normal (forward) and reversed peripheral nerve grafts demonstrated that the reversed polarity group exhibited an increase in cross-sectional area, axon count, and conduction velocity compared to the normal orientation group [27].

In our study, utilizing autologous facial nerve grafts from the opposite side, we observed no significant difference in nerve repair or functional recovery based on the suturing direction. Both methods demonstrated similar effects on nerve regeneration in the facial nerve axotomy animal model. Behavioral experiments indicated signs of recovery in both groups, with no significant differences between forward and reverse suturing techniques. While the number of axons decreased at 4 weeks, there was a partial recovery by 8 weeks, with no notable differences between the two suturing directions. Additionally, the thickness of myelinated fibers decreased at 8 weeks, showing no significant difference between the two methods. Overall, both forward and reverse suturing were effective for nerve regeneration, and no significant disparities were found in microstructural aspects. Both techniques exhibited similar trends in nerve regeneration and tissue-level recovery.

In terms of previous forward and reverse suturing studies, there are six studies in total: four using the sciatic nerve [25,27,28,29], one using the tibial nerve [31], and one using the common peroneal nerve [30]. However, all of these grafted nerves were sensory rather than motor nerves. Moreover, the size of the grafted nerves in these studies differed from that of the damaged facial nerve, complicating precise nerve reconnections. In contrast, our study offers two key advantages: first, the grafted nerve was the facial nerve, a motor nerve, similar to the damaged one; second, the size of the grafted nerve matched that of the damaged facial nerve. Based on the results of this study, we hypothesize that there would be no differences in motor function depending on the direction of nerve transmission, suggesting that effective nerve transmission is achievable regardless of suturing direction. However, further anatomical and physiological research is essential to elucidate the underlying mechanisms.

In addition to traditional morphological assessments, non-invasive imaging techniques like diffusion tensor imaging (DTI) offer promising alternatives for evaluating nerve regeneration. DTI, a type of magnetic resonance imaging (MRI), measures the anisotropy of water diffusion and enables the visualization of nerve integrity. This technology has demonstrated efficacy in visualizing axonal structures in both central and peripheral nervous systems, in both ex vivo and in vivo environments [32]. Applying DTI in future studies could allow a longitudinal observation of nerve regeneration and provide valuable insights into recovery dynamics, with the potential to correlate diffusion tensor metrics with conventional morphometric measurements. The limitations of our study include a reliance on high-magnification histological techniques, which restrict broader visual assessment and require invasive tissue sampling. Additionally, the longitudinal tracking of nerve regeneration was not feasible within the current methodology. Future studies could incorporate non-invasive techniques such as DTI, allowing for the real-time monitoring of nerve regeneration over time. Incorporating this technology would enable analysis of axonal structural integrity without the need for additional surgical interventions.

## 5. Conclusions

This study investigated the influence of nerve suturing direction (forward vs. reverse) on nerve regeneration outcomes through behavioral tests and histological analyses. Our findings indicated no significant impact of suture direction on nerve regeneration or functional recovery, as there were no substantial differences observed between the forward and reverse suturing methods. These results suggest that the orientation of suturing in nerve grafting may not play a critical role in the regeneration process. Future research should aim to explore the molecular mechanisms underlying nerve regeneration in greater depth, employing extended observation periods to enhance our understanding of this complex process.

## Figures and Tables

**Figure 1 biomedicines-12-02531-f001:**
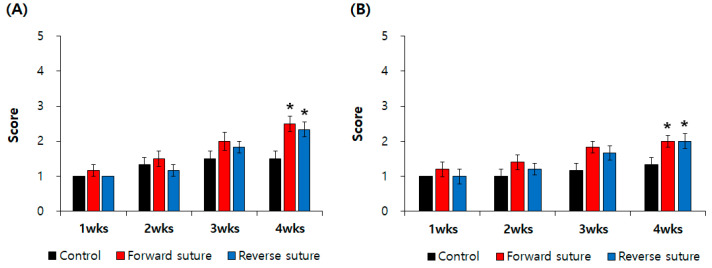
The control group (*n* = 6), the forward suture group (*n* = 6), and the reverse suture group (*n* = 6) underwent behavioral assessments using (**A**) whisker movement (vibrissae muscle) and (**B**) eyelid blink reflex tests at 1, 2, 3, and 4 weeks post-axotomy-induced facial nerve injury. Data are expressed as means ± SEM (* *p* < 0.05 vs. Control group). wks: weeks.

**Figure 2 biomedicines-12-02531-f002:**
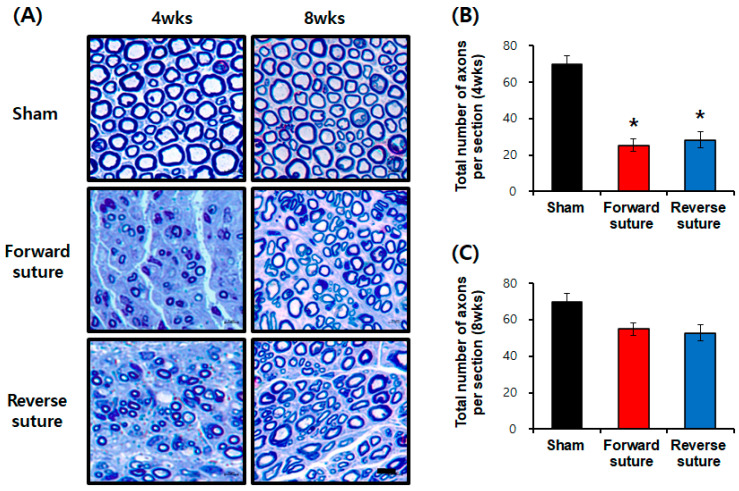
In the sham group (*n* = 3), forward suture group (*n* = 3), and reverse suture group (*n* = 3), facial nerve morphology was examined using toluidine blue staining at 4 and 8 weeks post-facial nerve axotomy to assess regeneration. (**A**) Toluidine blue staining images showing the morphological appearance of facial nerves in the sham, forward, and reverse suture groups at 4 and 8 weeks post-injury. (**B**) Quantitative analysis of axonal counts at 4 weeks post-injury, showing significantly more severe nerve damage in both suture groups compared to the sham group. (**C**) Quantitative analysis of axonal counts at 8 weeks post-injury, indicating no significant difference between suture groups and the sham group, suggesting morphological recovery in both groups. Scale bar: 5 μm. Data are expressed as means ± SEM (* *p* < 0.05 vs. sham group). wks; weeks.

**Figure 3 biomedicines-12-02531-f003:**
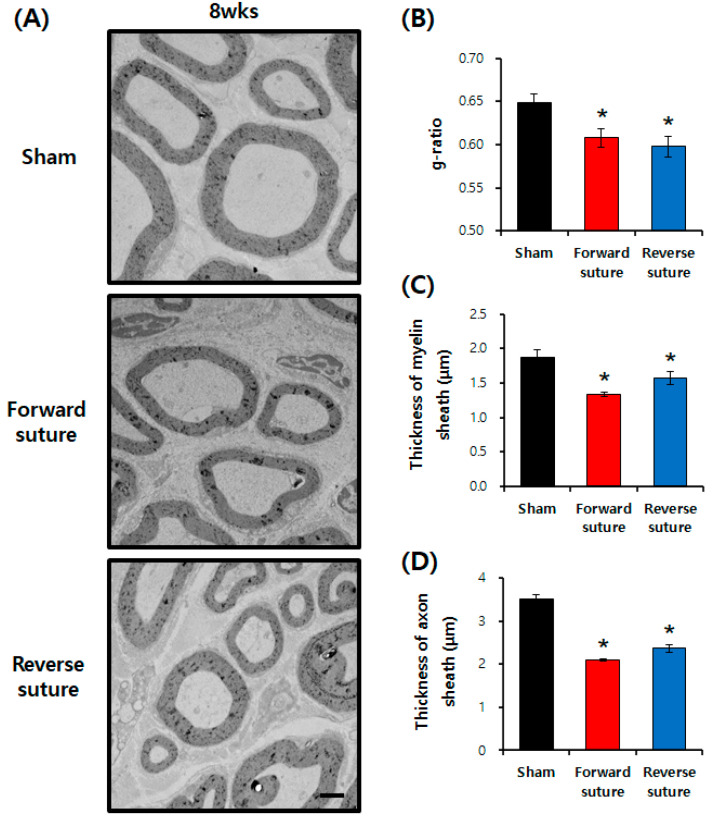
In the sham group (*n* = 3), forward suture group (*n* = 3), and reverse suture group (*n* = 3), facial nerve morphology was examined using transmission electron microscopy at 4 and 8 weeks post-facial nerve axotomy to assess regeneration. (**A**) TEM images showing the morphology of myelin and axon sheaths in sham, forward, and reverse suture groups. (**B**) g-ratio analysis assessing the degree of myelination recovery, with significantly lower g-ratios in the forward and reverse suture groups compared to the sham group. (**C**) Quantitative analysis of myelin sheath thickness at 8 weeks, demonstrating a significant reduction in both suture groups compared to the sham group. (**D**) Quantitative analysis of axon sheath thickness at 8 weeks, also showing a significant reduction in both suture groups relative to the sham group. Scale bar: 2 μm. Data are expressed as means ± SEM (* *p* < 0.05 vs. sham group). wks; weeks.

## Data Availability

The original contributions presented in the study are included in the article, further inquiries can be directed to the corresponding authors.

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
