# Peer review of "Effect of Forward and Reverse Suturing on Nerve Regeneration Following Facial Nerve Axotomy"

_biomedicines, 2024, doi:10.3390/biomedicines12112531_

Round 1

Reviewer 1 Report

Comments and Suggestions for Authors

In this manuscript, the authors examined how the direction of nerve suturing (forward vs. reverse) affects nerve regeneration after facial nerve injury in male Sprague-Dawley rats. Following facial nerve axotomy, autografts were sutured in both directions, and behavioral tests (whisker reflex and eye blink) were performed alongside histological analyses (Toluidine Blue staining and transmission electron microscopy). Results showed that both groups exhibited recovery signs at 3-4 weeks, with no significant differences in functional outcomes. Histological evaluations indicated partial recovery at 8 weeks in both groups, with a decrease in axon numbers at 4 weeks. TEM revealed myelin fiber thickness restoration in both groups, although still thinner than normal. Overall, the study concluded that suturing direction does not significantly affect nerve regeneration or functional recovery.

The manuscript is well-written, but there are some issues that need to be addressed:

  1. The methods section and the figure legends do not specify the number of mice in each group.
  2. In Figure 2, the 8-week panel should include a sham group, along with a statistical analysis of myelination percentages.
  3. For Figure 3, the electron microscopy results should include more representative images with larger fields of view. The statistical results are too simplistic; additional statistics regarding the g-ratio, axon diameter, and myelination percentages should be included.

Author Response

First, we would like to thank the Reviewer for their thorough review and valuable feedback on our manuscript. We have organized our responses to the Reviewer’s comments in the attached Word file. Thank you for your attention and consideration.

Reviewer 2 Report

Comments and Suggestions for Authors

The abstract has been prepared in accordance with the authors guidelines. A stylistic note: "blue" in Toluidine blue may be formatted in small caps for consistency.

The manuscript is rigorously written. The primary issue with this article lies in the textual similarity to a previously published article by the same authors. The iThenticate report indicates a 45% text match with earlier publications, which is notably high, especially within the methods section. Authors are advised to revise the manuscript, focusing on rephrasing the methods to reduce similarity with prior work.

Line 105: please explain abbreviation PBS – phosphate buffered saline

Fig 2. Please explain the abbreviation wks or rather write the entire word on the image “weeks”

Lines 136-38. Please explain more precise how were these changes measured on toluidine-stained sections. Manually I assume? How many evaluators? Their experience? FOV?

The discussion is fairly written but it could still be improved. Authors should emphasise some less invasive methods for evaluation of peripheral nerve integrity and architecture in such cases. Diffusion tensor imaging is one of such. It is magnetic resonance imaging technique that is feasible within ex vivo and in vivo environment. I recommend to the authors that they refer the the following article and emphasise that future studies can less invasively evaluate nerve regeneration after facial nerve axotomy using diffusion tensor imaging. Furthermore, some correlations between diffusion tensors and morphometric assessment could be also beneficial in future studies. See and comment the reference: https://doi.org/10.3389/fphys.2023.1070227

Some limitations of the study should be addressed.

Author Response

(The authors gave the same response as above.)
